# Encapsulation of Hemp (*Cannabis sativa* L.) Essential Oils into Nanoemulsions for Potential Therapeutic Applications: Assessment of Cytotoxicological Profiles

**DOI:** 10.3390/molecules28186479

**Published:** 2023-09-07

**Authors:** Cristina Aguzzi, Diego Romano Perinelli, Marco Cespi, Laura Zeppa, Eugenia Mazzara, Filippo Maggi, Riccardo Petrelli, Giulia Bonacucina, Massimo Nabissi

**Affiliations:** 1Department of Experimental Medicine, School of Pharmacy, University of Camerino, Via Madonna delle Carceri 9, 62032 Camerino, Italy; cristina.aguzzi@unicam.it (C.A.); laura.zeppa@unicam.it (L.Z.); 2Chemistry Interdisciplinary Project (ChIP) Research Center, School of Pharmacy, University of Camerino, Via Madonna delle Carceri, 62032 Camerino, Italy; diego.perinelli@unicam.it (D.R.P.); marco.cespi@unicam.it (M.C.); eugenia.mazzara@unicam.it (E.M.); filippo.maggi@unicam.it (F.M.); riccardo.petrelli@unicam.it (R.P.)

**Keywords:** MTT assay, cell toxicity, cytokine gene expression, nanoencapsulation, dynamic light scattering, topical formulation

## Abstract

Industrial hemp (*Cannabis sativa* L.), due to its bioactive compounds (terpenes and cannabinoids), has gained increasing interest in different fields, including for medical purposes. The evaluation of the safety profile of hemp essential oil (EO) and its encapsulated form (nanoemulsion, NE) is a relevant aspect for potential therapeutic applications. This study aimed to evaluate the toxicological effect of hemp EOs and NEs from cultivars Carmagnola CS and Uso 31 on three cell lines selected as models for topical and inhalant administration, by evaluating the cytotoxicity and the cytokine expression profiles. Results show that EOs and their NEs have comparable cytotoxicity, if considering the quantity of EO present in the NE. Moreover, cells treated with EOs and NEs showed, in most of the cases, lower levels of proinflammatory cytokines compared to Etoposide used as a positive control, and the basal level of inflammatory cytokines was not altered, suggesting a safety profile of hemp EOs and their NEs to support their use for medical applications.

## 1. Introduction

In recent years, the essential oil (EO) from industrial hemp (*Cannabis sativa* L.) has received great scientific and commercial attention as a valuable resource that could be potentially exploited in different fields [1,2]. Hemp EO is predominantly composed of monoterpenes (α-pinene, myrcene, and terpinolene) and sesquiterpenes (α-humulene and caryophyllene), with a lower content of the cannabinoid fraction, among which cannabidiol (CBD) generally represents the main component [3,4]. Therefore, hemp EO configures as a nonpsychotropic natural product endowed with plenty of biological activities (e.g., insecticidal, parasiticidal, antimicrobial, and anti-inflammatory) which make it attractive for agrochemical, cosmeceutical, nutraceutical, and pharmaceutical applications [5,6,7]. Regarding human health, hemp EO is reported to exert several benefits related to the pharmacological actions of the single compounds or the entire phytocomplex (cannabinoids and terpenes), which include immunomodulatory, anticonvulsant, anxiolytic, neuroprotective, spasmolytic, immunomodulatory, anti-inflammatory, and anticancer effects [8,9,10]. The use of hemp EO in topical formulations as a functional ingredient is constantly increasing [11]. Indeed, hemp EO has been largely investigated for its beneficial effect on inflammatory skin diseases, such as acne, seborrhea, dermatitis, and psoriasis [12,13]. As a primary anatomical barrier, skin is continuously subjected to a variety of external physical stimuli, responding through the modulation of specific pathways. Keratinocytes represent the epidermis’s main cellular population, and they are involved in activating and maintaining inflammation and immunological responses [14,15]. Fibroblasts, in the epidermis, are involved in the wound healing process, and act synergistically with keratinocytes in maintaining homeostatic conditions [16]. Keratinocytes and fibroblasts release several proinflammatory mediators, including interleukin-1 beta (IL-1 beta), IL-6, IL-8, and tumor necrosis factor-alpha (TNF alpha), under inflammatory stimuli. Traditionally, several plants’ EOs have long been used in medicine as expectorant, bronchodilator, anti-inflammatory, antiviral, antibacterial, and antiseptic agents, mainly in the treatment of upper respiratory system issues [17,18]. More recently, CBD-enriched extracts have been demonstrated to have an immunomodulatory effect on pulmonary cell lines due to a downregulation of the cytokine storm or a modulation of gene expression, which can be beneficial for the treatment of lung inflammatory chronic diseases [19,20]. Nonetheless, due to the poor water solubility and the possibility of chemical degradation and biological inactivation of hemp EO into the environment, effective and safe formulations for the administration and delivery to the target are required. An efficient strategy to overcome these issues is offered by the encapsulation into nanoemulsions (NEs) [21,22]. According to these premises, the main objective of this study was to compare the safety profiles of pure hemp EOs and their NEs in order to assess the effect of nanoencapsulation on cytotoxicity and proinflammatory cytokines. Monoecious variety Uso 31 and male inflorescences of the dioecious variety Carmagnola CS usually represent waste products of hemp processing, and for this reason, they were chosen for this work in order to exploit their still underestimated potential. These studies were carried out in human keratinocytes, fibroblasts, and bronchial cell lines, selected as a model for topical administration on the skin or for pulmonary vehiculation of hemp EO-based nanoformulations.

## 2. Results

### 2.1. EO Chemical Composition

Both hemp inflorescences provided a comparable yield in terms of the amount of the EO obtained after the extraction procedure. Specifically, Uso 31 and Carmagnola CS yielded 0.18% and 0.20% *w*/*w* of EO on a dry-weight basis, respectively. The determined density values were also similar, being 0.884 and 0.896 g/mL for Uso 31 and Carmagnola CS EOs, respectively. Table 1 contains the main chemical constituents of the two EOs, in terms of relative abundance with respect to the whole composition, which is reported in Appendix A. In the case of Uso 31 EO, the predominant compound was the sesquiterpene (E)-caryophyllene (25.93%), followed by the monoterpene myrcene (15.28%). Other components present in significant percentages were especially α-pinene (11.07%), α-humulene (8.92%), caryophyllene oxide (7.23%), and (E)-β-ocimene (6.71%). On the other hand, the monoterpene myrcene dominated in Carmagnola CS EO, accounting for 37.57% of the chemical profile. The second most abundant constituent was (E)-caryophyllene (16.99%), followed by α-pinene (13.45%) and α-humulene (6.11%). The content of the other detected compounds was under 6.0%. Specifically, compounds of the cannabinoid fraction (including CBD) are present in the EOs only at trace levels.

### 2.2. Characterization and Physical Stability over Time of C. sativa EO-Based NEs

A preliminary screening was carried out to select the best *C. sativa* EO-based NE formulation in terms of total oil (EO plus ethyl oleate) incorporation (%) and surfactant (polysorbate 80) concentration (%) and to have a mean droplet size (Z-average, diameter) and PDI lower than 200 nm and 0.3, respectively. The addition of ethyl oleate in the formulation was required for a better solubilization of the *C. sativa* EO, thereby ensuring a longer stability time as reported in a previous work [21]. As illustrated in Figure 1A, the particle size distribution of the prepared 5% *w*/*w* Uso 31 EO-based NE and 6% *w*/*w* Carmagnola CS EO-based NE is below 1000 nm, with the maximum centered at around 100–150 nm. A slightly larger width of the droplet distribution (PDI values 0.27 vs. 0.24, respectively) and a slightly higher Z-average value (131 nm vs. 117 nm) were determined for Carmagnola CS EO-based NE with respect to the Uso 31-based NE, thereby reflecting the different percentage of the incorporated EO (6% *w*/*w* vs. 5% *w*/*w*). Both NEs did not display any macroscopic physical instability phenomena (e.g., creaming or phase separation) over the observation time up to six months. Moreover, Z-average and PDI values remained for both NEs in the range of 110–160 nm and 0.24–0.27, respectively, indicating the good stability at room temperature of the nanosystems, without appreciable variation in the droplet size of the internal oil phase (Figure 1B,C).

### 2.3. Evaluation of C. sativa EO-Based NEs’ Safety Profile in Human Cell Lines

The safety profile of *C. sativa* EO-based NEs was investigated in human keratinocytes (HaCaT), human fibroblasts (NHF A12), and human bronchial cell (HBEpC) lines, to assess at which noncytotoxic concentrations these nanoformulations can be administered topically on the skin or by inhalation. As such, cells were treated with different dilutions (% *v*/*v*) of *C. sativa* EO-based NEs and pure EOs for comparison. Results show that HBEpC cells were more sensitive, showing lower viability values (%) than HaCaT and NHF A12 both for the pure EOs and NEs formulations (Figure 2). Notably, *C. sativa* EO-based NEs induced a lower cytotoxic effect compared to *C. sativa* EOs for all tested cell lines, since the two NEs contain only 5% or 6% *w*/*w* EO with respect to their total composition, while the main constituent is water. A much lower cytotoxic effect than *C. sativa* EO-based NEs was observed for control NEs against all tested cell lines. Specifically, the residual effect on cell viability for control (CTRL) NEs can be mainly attributed to the presence of polysorbate 80 used as an emulsifier in the formulations. 

A thorough comparison between *C. sativa* EO-based NEs and the corresponding EOs can be performed by calculating the IC_50_ values (Table 2). Similar IC_50_ values were determined for the two *C. sativa* EOs against the three tested cell lines, highlighting that the two hemp varieties exert the same cytotoxic effect. Regarding *C. sativa* NE formulations, their IC_50_ values were much higher than those of the corresponding pure EOs against the same cell lines, as expected from the composition of the formulation (containing the 5% or 6% *w*/*w* of EOs with respect to the total weight). Nevertheless, for a direct comparison of the cytotoxic effect exerted by *C. sativa* NEs with that of pure EOs, the IC_50_ values of NEs after normalization by the percentage of EO contained in the formulation are reported in brackets in Table 2. The normalized IC_50_ values for NEs are slightly lower but still comparable to those calculated for pure EOs, suggesting that the incorporation of the EOs in the formulations does not markedly affect the cytotoxic profile of *C. sativa* EOs. On the other hand, the slight difference in IC_50_ of the *C. sativa* NEs can be ascribed to the different percentages of the other two components, polysorbate 80 and ethyl oleate. Indeed, Carmagnola CS NE, formulated with 3% *w*/*w* of polysorbate 80 and 2% *w*/*w* of ethyl oleate, showed lower IC_50_ values than Uso 31 NE, which instead contained 2% *w*/*w* of polysorbate 80 and 1% *w*/*w* of ethyl oleate. The possible effect exerted by polysorbate 80 and ethyl oleate on cytotoxicity is also evident by comparing the IC_50_ values of the two CTRL NEs.

### 2.4. Evaluation of C. sativa EO-Based NEs’ Effect on Inflammatory Condition of Human Cell Lines

According to the results obtained from the cytotoxic assay, HaCaT, NHF A12, and HBEpC cells were treated for 24 h with 0.0039% (% *v*/*v*) of *C. sativa* EOs and 0.015% (% *v*/*v*) of *C. sativa* EO-based NEs and their respective control. Cells were also treated with ETO 2.5 µM used as a positive control to induce inflammation. The gene expression of proinflammatory cytokines, such as IL-1 beta, IL-6, TNF-alpha, and STAT3, were measured by RT-PCR. From a general point of view, *C. sativa* EOs and their NEs did not induce an inflammatory state compared to ETO in all cell types used. In the HaCaT cell line, Carmagnola CS EO and Uso 31 EO slightly increased IL-1 beta expression, and the effect is partially reduced for *C. sativa* EO-based NEs. *C. sativa* EO-based NEs can revert the basal condition as demonstrated especially for IL-6 and TNF-alpha in the HaCaT model. Regarding STAT3, *C. sativa* EOs, in the encapsulated and nonencapsulated forms, did not alter their basal expression (Figure 3).

In NHF A12, among all proinflammatory genes analyzed, the larger reduction in expression was observed for TNF-alpha when cells were treated with Uso 31 EO-based NE. Other statistically significant reductions in comparison to the untreated cells and the pure EO were observed for Uso 31 EO-based NE in the case of STAT3 and IL-6 expression. Moreover, a statistically significant reduction of IL-1β expression was observed for Carmagnola CS EO-based NE in comparison to the pure EO (Figure 4).

Similar results were obtained for the human bronchial HBEpC cell line, in which the basal expression was not hampered by the treatments, but, additionally, IL-6 expression was reduced by Carmagnola CS EO-based NE (Figure 5).

## 3. Discussion

The use of *C. sativa* as a medicinal plant has a long history in the field of traditional medicine [23]. Indeed, the medical interest around this botanical species has renewed in the last century thanks to the discovery of phytocannabinoids, which are among the main constituents of cannabis, to which are ascribed the well-known anti-inflammatory, analgesic, myorelaxant, and psychoactive effects [24]. Most of the *C. sativa* varieties, known as “industrial hemp” and available for cultivation, are poor in the cannabinoids fraction, but they contain a noticeable fraction of terpenes and terpenoids. These compounds are also recognized to exert several pharmacological activities as antifungal, antiviral, anticancer, anti-inflammatory, antihyperglycemic, antiparasitic, and antioxidant agents [25]. In this work, Carmagnola CS and Uso 31 were selected as “industrial hemp” varieties to extract the EOs, which contain more than 95% (according to GC analysis) of the terpenes/terpenoids fraction, and only traces of phytocannabinoids. NEs have been formulated to encapsulate these EOs with the aim of investigating the toxicological profiles and understanding the impact of their encapsulation in terms of inflammatory cytokine expression.

Regarding the chemical composition, quali-quantitative differences in the relative abundances of hemp EOs’ main constituents can occur due to several parameters affecting EO profile, such as genetics, cultivation practice, harvesting, storage, and drying conditions [26]. Few studies can be found in the literature regarding the chemical composition of Uso 31 hemp EO [3,27]. By comparing our research work with those by Pieracci et al. [3] and Ascrizzi et al. [27], component content variability could depend especially on the different plant material status (fresh in our study, and dry in that by [3]) and distillation method (the innovative MAE here, and the conventional HD in both the Ascrizzi et al. [27] and Pieracci et al. [3] papers). Similar considerations can be made concerning the EO from Carmagnola CS male inflorescences. In our previous work on the steam-distilled EO of the same variety coming from the same hemp farm [21], sesquiterpenes, particularly (*E*)-caryophyllene, were predominant, in contrast to the current study. Such differences could be a consequence of the diverse harvesting periods and extraction techniques.

The encapsulation of EOs with potential therapeutic applications could contribute to the development of formulations with a practical use by increasing their stability and bioavailability, and by helping to overcome the drawbacks associated with the occurrence of side effects related to therapy. The incorporation of Carmagnola CS and Uso 31 EOs in the formulated NEs resulted in physically stable systems with a mean droplet size in the nanometric range and a narrow size distribution. After encapsulation into NEs, both EOs showed similar toxicological profiles, compared to pure EOs, toward the investigated cell lines (HaCaT, NHF A12, and HBEpC), as revealed by the normalized IC_50_ values. By considering the not normalized IC_50_ values directly determined from MTT assay, NEs revealed a safety profile from a toxicological point of view. Indeed, the calculated IC_50_ values for NEs (containing 5–6% *w*/*w* of EO) were much higher than those from pure EOs, despite being influenced by the amount of EOs and NE composition in terms of content of surfactant (polysorbate 80) and cosolvent (ethyl oleate). To corroborate the safety profile of EOs and their NEs, in light of their potential applications as therapeutics, analysis of inflammatory gene expression profiles on the same cell lines was performed. The results, reported here, are coherent with those previously obtained for *C. sativa* EOs (Felina 32 and male Carmagnola CS varieties), for which an anti-inflammatory effect on the tested human cell lines was observed [21]. In this study, we confirmed that the EOs and their NEs at the doses tested do not induce an inflammatory condition, in terms of inflammatory gene expression, in comparison to ETO. However, the three cell lines showed different results with respect to EOs and NEs. In general, the encapsulation of the EO leads to a better modulation of the analyzed cytokine levels, as can be observed for Uso 31 EO-based NE, particularly towards the HaCaT and NHF A12 cell lines, or for *C. sativa* EO-based NEs towards the HBEpC cell line. As already observed in previous studies regarding other EOs [28], the encapsulation into nanosystems could potentially improve the safety profile in human cell lines also in terms of the cytokine expression profile.

## 4. Materials and Methods

### 4.1. Plant Material

The monoecious inflorescences of Uso 31 and the male inflorescences of Carmagnola CS hemp varieties were grown by La Biologica farm (Fiuminata, central Italy, 43°10′40″ N, 12°56′59″ E, 451 m a.s.l.) and harvested at the beginning of August 2020. Then, all the hemp biomass was frozen until use. The moisture content of plant material was evaluated by using a hot air oven (BINDER GmbH, Tuttlingen, Germany) at 105 °C for 24 h, and the average water content was 70.2 and 80.3% for Uso 31 and Carmagnola CS inflorescences, respectively.

### 4.2. Microwave-Assisted Extraction (MAE)

Uso 31 and Carmagnola CS male inflorescences were processed using Milestone ETHOS X microwave equipment (Milestone, Sorisole, Italy), used at the maximum irradiation power of 1800 W for 100 min. About 2 kg of frozen hemp inflorescences, defrosted for 30 min, were placed in a glass reactor inside the microwave apparatus, equipped with a stainless steel Clevenger. The volatile compounds of interest were collected in the form of an EO, due to a circulating cold water flow, maintained at 8 °C by a Chiller Smart H150-2100S by Labtech srl (Sorisole, Bergamo, Italy). The obtained EOs were collected by the collection burette after separation from water and kept in a sealed vial at 4 °C until further analysis. The EO yield was measured on dry matter (*w*/*w*). The density of the two hemp EOs was calculated at 20 °C with an oscillating U-tube density meter (DA-100M, Mettler Toledo, Greifensee, Switzerland).

### 4.3. EO GC-MS Characterization

The chemical composition of the two hemp EOs was assessed according to a previously published GC-MS method [29], using an Agilent 8890 (GC) equipped with an Agilent 5977B Mass Spectrometer (Santa Clara, CA, USA). Compounds were identified by comparing mass spectra with those present in commercial libraries, namely, ADAMS [30], NIST17 [31], and FFNS C3 [32].

### 4.4. C. sativa EO-Based Nanoemulsions (NEs) Formulation and Characterization

NEs were prepared through a high-pressure homogenization method. Firstly, an emulsified system was achieved by adding the required amount of EO, previously solubilized in ethyl oleate, into a polysorbate 80 aqueous solution under high-speed stirring (Ultra Turrax T25 basic, IKA^®^ Werke GmbH & Co. KG, Staufen, Germany) applied for 5 min at 9500 rpm. Then, the emulsions were homogenized at the pressure of 130 MPa using a French Pressure Cell Press (American Instrument Company, AMINCO, Silver Spring, MY, USA) apparatus for 4 cycles. The final composition of the prepared NEs was 5% *w*/*w* EO, 2% *w*/*w* polysorbate 80, 1% ethyl oleate, and 92% distilled water for Uso 31 EO-based NE, and 6% *w*/*w* EO, 3% *w*/*w* polysorbate 80, 2% ethyl oleate, and 89% distilled water for Carmagnola EO-based NE. Control (CTRL) NEs without hemp EOs were prepared according to the same procedure. Droplet size (Z-average, diameter) and polydispersity index (PDI) were assessed by dynamic light scattering (DLS) (Zetasizer nanoS, Malvern, Worcestershire, UK) after preparation (T0) and at different time points (one month, T30; two months, T60; four months, T120; and six months, T180) to check the physical stability of the formulations. Analyses were performed in three replicates.

### 4.5. Cell Lines

Immortalized human keratinocytes cell line (HaCaT) and human fibroblasts (NHF A12), provided by IFO (Istituti Fisioterapici Ospitalieri, Rome, Italy), were cultured in Dulbecco’s modified Eagle’s medium (DMEM; Lonza Bioresearch, Basel, Switzerland), enriched with 10% fetal bovine serum (FBS, Thermo Fisher Scientific Inc., Waltham, MA, USA), 100 IU mL^−^^1^ penicillin/streptomycin, and 2 mM L-glutamine and kept at 37 °C with 5% CO_2_ and 95% humidity. Human bronchial epithelial cells (HBEpC), obtained from the surface epithelium of normal human bronchi (Sigma Aldrich, Milan, Italy), were grown in Bronchial Epithelial Cell Growth medium (Sigma Aldrich, Milan, Italy), following the manufacturing protocol. This medium was specifically designed to promote attachment, spreading, and proliferation of HBEpC cells in culture. This medium is serum-free, and it was fully supplemented with growth factors, trace elements, and antibiotics. Media were changed every 48 h until cells were 90% confluent. Cells were used between passages three and six. Aliquots of passage three were frozen in liquid nitrogen and cultured until passage six.

### 4.6. Cell Viability Assay

Cell lines (3 × 10^4^ cells/mL) were seeded in 96-well plates, in a final volume of 100 µL/well, and after one day of incubation, EOs, EO-based NEs, and respective NEs CTRL were added. At least six replicates in each experiment were used for each treatment. After 72 h, cell viability was assessed by adding 0.8 mg/mL of 3-[4,5-dimethylthiazol-2-yl]-2,5 diphenyl tetrazolium bromide (MTT) (Sigma-Aldrich, Milan, Italy) to the media. The absorbance of samples, solubilized in dimethyl sulfoxide (DMSO), against a background control (medium alone) was measured at 570 nm using a reader microliter plate (BioTek Instruments, Winooski, VT, USA).

### 4.7. RNA Isolation, Reverse Transcription and Quantitative Real-Time PCR, and TaqMan Array

Total RNA from cells treated with Etoposide (ETO), EOs, EO-based NEs, and respective controls was extracted with the RNeasy Mini Kit (Qiagen, Milan, Italy), and cDNA was synthesized using the iScript Advanced cDNA Synthesis Kit for RT-qPCR (Bio-Rad, Milan, Italy), according to manufacturer’s protocol. Quantitative real-time polymerase chain reactions (qRT-PCR) were performed with TaqMan^®^ Array, containing 14 genes and 2 assays to candidate endogenous control genes, which was purchased (Thermo Fisher, Grand Island, NY, USA) and used to evaluate the safety profile of each treatment compared to ETO-treated cells used as a positive control. Measurement of housekeeping gene glyceraldehydes-3-phosphate dehydrogenase (*GAPDH*) on the samples was used to normalize mRNA content. The gene expression levels of treated cell lines were expressed as normalized fold to untreated cells. For detection, the iQ5 Multicolor Real-Time PCR Detection System (Bio-Rad, Milan, Italy) was used. The PCR parameters were 10 min at 95 °C followed by 40 cycles at 95 °C for 15 s and 60 °C for 40 s. Target gene levels were calculated by the 2^−ΔΔCt^ method.

### 4.8. Statistical Analysis

The presented data represent the mean with standard deviation (SD) of at least 3 independent experiments. The statistical significance was determined by Student’s *t*-test and by one-way ANOVA and two-way ANOVA with Bonferroni’s post-test. Determination and statistical analysis of the levels of the inhibitory concentration causing 50% of cell death (IC_50_) were performed using Prism 5.0a (GraphPad Software, San Diego, CA, USA).

## 5. Conclusions

Results from this study highlighted the cytotoxicological profiles of hemp (*C. sativa*) EOs, specifically the Carmagnola CS and Uso 31 varieties, and their formulated NEs on three different cell lines, human keratinocytes (HaCaT), fibroblasts (NHF A12), and bronchial epithelial cells (HBEpC). The encapsulation of EOs into NEs does markedly affect the safety profiles of EOs, which are comparable among all the tested cell lines. The prepared NEs were stable over time for at least 4 months, and they showed IC_50_ values apparently higher than those from the corresponding pure EOs, but formulation parameters (EO, surfactant, and cosolvent concentration) can slightly affect cytotoxicity, at least under the investigated experimental conditions. Cytokine expression analysis showed a safe profile, since cytokine levels were almost lower than those induced by the positive control ETO and, in most of the cases, comparable to the basal level (untreated cells), although some differences have been observed among the expression of the investigated cytokines (IL-1 beta, IL-6, TNF-alpha, and STAT3). Overall, the preliminary results of the present study regarding the cellular toxicity and cytokine expression support the possible safe use of hemp EO nanoformulations such as topically or respiratory-administered NEs for therapeutic applications.

## Figures and Tables

**Figure 1 molecules-28-06479-f001:**
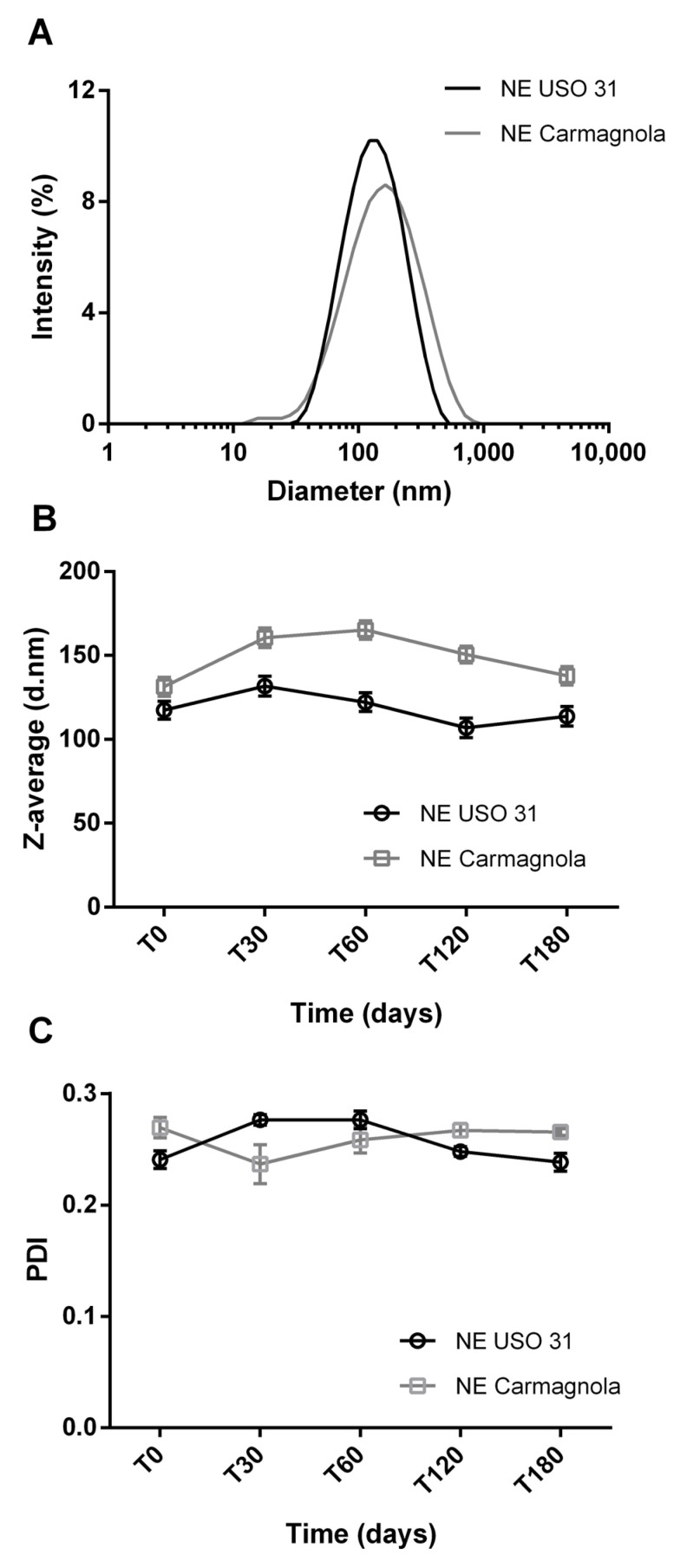
Particle size distribution by intensity (%) as recorded by DLS for Uso 31 and Carmagnola CS EO-based NEs after preparation (**A**) and variation of Z-average (diameter, d., nm) (**B**) and polydispersity index (PDI) (**C**) values as measured over time up to six months.

**Figure 2 molecules-28-06479-f002:**
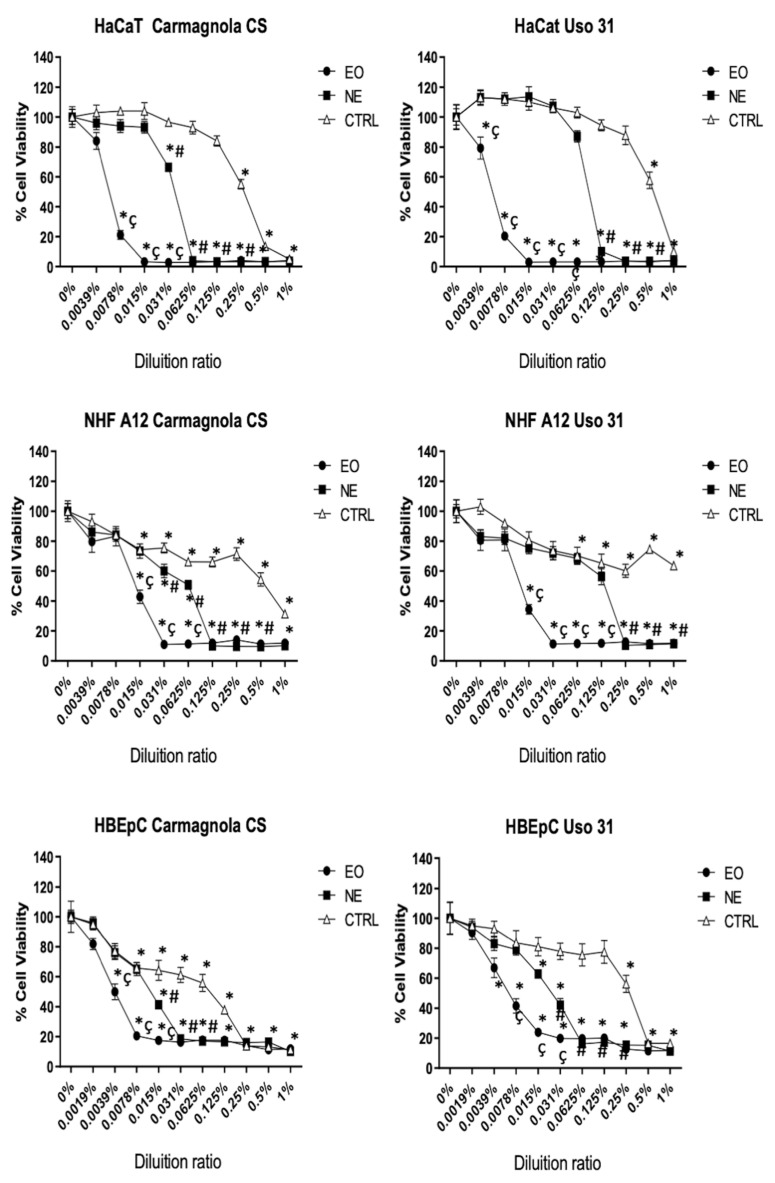
Cell cytotoxicity of Uso 31 and Carmagnola CS EOs and their NEs in HaCaT, NHF A12, and HBEpC cell lines was determined by MTT assay. All cell lines were treated with different dilutions, and cell viability was evaluated after 72 h post-treatment. * *p* < 0.05 vs. vehicle; ^#^ *p* < 0.05 *C. sativa* EO-based NEs vs. CTRL; ^ç^
*p* < 0.05 *C. sativa* EO-based NEs vs. *C. sativa* EO.

**Figure 3 molecules-28-06479-f003:**
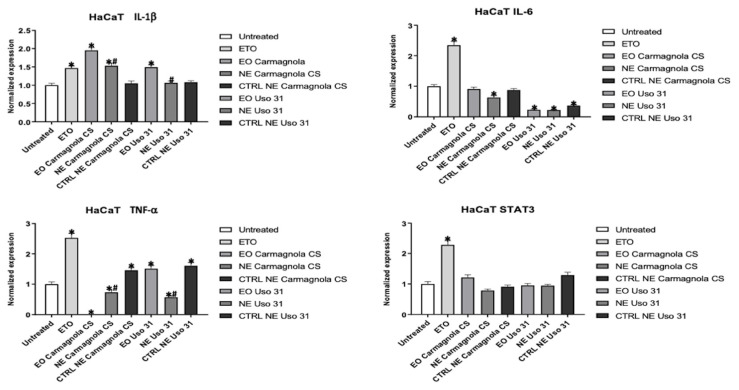
Effect of Uso 31 and Carmagnola CS EO-based NEs on the inflammatory state was evaluated in HaCaT cell line. IL-1 beta, IL-6, TNF-alpha, and STAT3 mRNA levels were determined by RT-PCR. Cells were treated with 0.0039% (% *v*/*v*) of *C. sativa* EOs and 0.015% (% *v*/*v*) of *C. sativa* EO-based NEs and their control, compared with cells treated with ETO 2.5 µM used as a positive control, and evaluated after 24 h. * *p* < 0.05 vs. untreated; ^#^
*p* < 0.05 *C. sativa* EO-based NEs vs. *C. sativa* EO.

**Figure 4 molecules-28-06479-f004:**
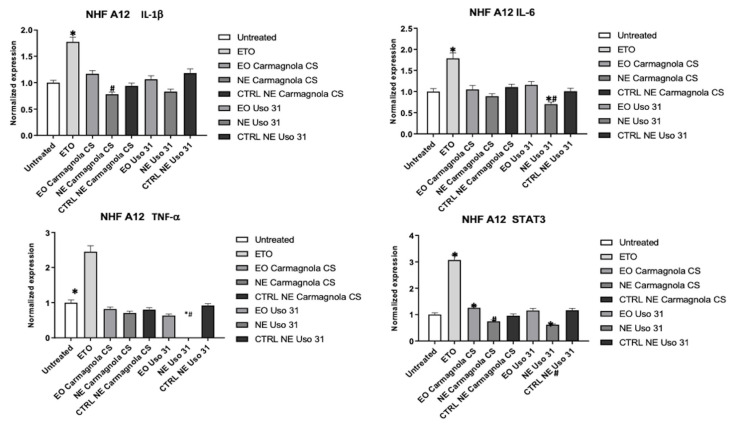
Effect of Uso 31 and Carmagnola CS EO-based NEs on inflammatory state was evaluated in NHF A12 cell line. IL-1 beta, IL-6, TNF-alpha, and STAT3 mRNA levels were determined by RT-PCR. Cells were treated with 0.0039% (% *v*/*v*) of *C. sativa* EOs and 0.015% (% *v*/*v*) of *C. sativa* EO-based NEs and their control, compared with cells treated with ETO 2.5 µM used as positive control, and evaluated after 24 h. * *p* < 0.05 vs. untreated; ^#^
*p* < 0.05 *C. sativa* EO-based NEs vs. *C. sativa* EO.

**Figure 5 molecules-28-06479-f005:**
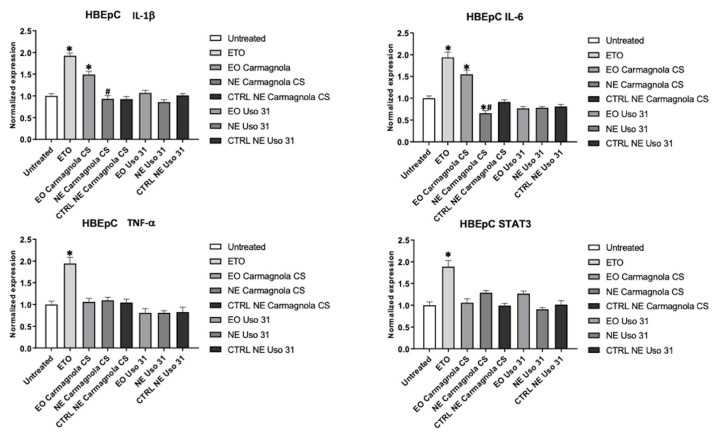
Effect of Uso 31 and Carmagnola CS EO-based NEs on inflammatory state was evaluated in HBEpC cell line. IL-1 beta, IL-6, TNF-alpha, and STAT3 mRNA levels were determined by RT-PCR. Cells were treated with 0.0039% (% *v*/*v*) of *C. sativa* EOs and 0.015% (% *v*/*v*) of *C. sativa* EO-based NEs and their control, compared with cells treated with ETO 2.5 µM used as positive control, and evaluated after 24 h. * *p* < 0.05 vs. untreated; ^#^
*p* < 0.05 *C. sativa* EO-based NEs vs. *C. sativa* EO.

**Table 1 molecules-28-06479-t001:** Main chemical constituents of Uso 31 and Carmagnola CS EOs.

Component ^a^	RI Calc. ^b^	RI Lit ^c^	% Uso 31	% Carmagnola CS
α-pinene	926	932	11.07	13.45
β-pinene	968	974	3.59	5.45
myrcene	989	988	15.28	37.57
limonene	1025	1024	1.59	5.37
(*E*)-β-ocimene	1047	1044	6.71	1.79
terpinolene	1085	1086	5.46	2.97
(*E*)-caryophyllene	1409	1417	25.93	16.99
α-humulene	1443	1452	8.92	6.11
caryophyllene oxide	1571	1582	7.23	2.95
**Total identified (%)**			99.03	97.72
Monoterpene hydrocarbons (%)			44.24	67.27
Oxygenated monoterpenes (%)				0.43
Sesquiterpene hydrocarbons (%)			45.31	26.18
Oxygenated sesquiterpenes (%)			9.04	3.58
Cannabinoids (%)			0.43	0.23

^a^ Order of compounds according to their elution from HP-5MS column. ^b^ Linear retention index calculated with a mixture of n-alkanes (C_8_–C_30_) with respect to HP-5MS column. ^c^ Retention index for nonpolar columns taken from ADAMS library.

**Table 2 molecules-28-06479-t002:** Calculated IC_50_ values (mg mL^−1^) for Carmagnola CS and Uso 31 EOs and their NEs in comparison with control NE (CTRL NE) against different cell lines (HaCaT, NHF A12, and HBEpC).

	IC50 mg mL−1		
	HaCaT	NHF A12	HBEpC
Carmagnola CS EO	0.052 ± 0.002	0.119 ± 0.009	0.034 ± 0.002
Carmagnola CS NE	0.354 ± 0.018 (0.021 ± 0.001)	0.448 ± 0.020 (0.023 ± 0.001)	0.154 ± 0.010 (0.009 ± 0.001)
CTRL NE	2.606 ± 0.050	9.666 ± 0.600	1.221 ± 0.500

Uso 31 EO	0.049 ± 0.002	0.103 ± 0.008	0.041 ± 0.003
Uso 31 NE	0.857 ± 0.030 (0.043 ± 0.002)	1.505 ± 0.058 (0.075 ± 0.003)	0.245 ± 0.010 (0.012 ± 0.001)
CTRL NE	5.388 ± 0.100	>10.04	3.255 ± 0.090

IC_50_ values of NEs after normalization by the percentage of EO contained in the formulation are reported in the brackets.

## Data Availability

The data presented in this study are available on request from the corresponding author.

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
