# Peer review of "Encapsulation of Hemp (Cannabis sativa L.) Essential Oils into Nanoemulsions for Potential Therapeutic Applications: Assessment of Cytotoxicological Profiles"

_molecules, 2023, doi:10.3390/molecules28186479_

Round 1

Reviewer 1 Report

This paper is well-written and contributes to potential therapeutic applications. Through fabrication into nanoemulsions let hemp (Cannabis sativa L.) essential oils have a new ability and decrease the cytotoxicological.

But this article has a few questions.

1: The title talks about the potential therapeutic, would you please add some experiment results related to therapeutic?

2: How can it be proved that it is encapsulated, and what is the rate of encapsulation, is there any hemp oil that can not loaded into this drug delivery system?

3: About the size, Zeta and PDI, Is there any significant difference between them, please mark it out.

4: In the cell cytotoxicological results, you were using many different types of cells. Could you please explain in detail why the toxicity is reduced after making nanoemulsion?

In your article, you said it attributed to the presence of polysorbate 80. Could you please add some date or published paper to prove it?

Another question, what is the crtl? If it is a control group, why does it also have toxicity?

5: About the Figure 3 Hacat IL-1β group. EO and NE group cause the high level of IL-1β, how to explain it?

6: The workload of this article is a bit small, could you add some more data?

Thanks

Author Response

Reviewer 1

This paper is well-written and contributes to potential therapeutic applications. Through fabrication into nanoemulsions let hemp (Cannabis sativa L.) essential oils have a new ability and decrease the cytotoxicological.

The authors thank the reviewer for the positive general comment about the manuscript.

But this article has a few questions.

1: The title talks about the potential therapeutic, would you please add some experiment results related to therapeutic?

The aim of our study was to perform a preliminary assessment of the cytotoxicological profile of two hemp EOs encapsulated into nanoemulsions and to understand whether the formulation of these EOs into nanoemulsions can affect the cytotoxicity of the EOs themselves on three selected cell lines as human keratinocytes (HaCaT), human fibroblasts (NHF A12), and human bronchial cell (HBEpC) as model for the topical applications and the respiratory route. The sentence “potential” was added since different cytokines modulated by EOs are involved in acute and chronic skin and lung inflammatory diseases. However, we agree with the reviewer and although the evaluation of the therapeutic applications of these formulations goes beyond the scope of the work, further experiments focused on specific skin or lung disease models will be considered for further studies. Some changes have been made in the Introduction and conclusion sections to specify better the aim of the work.

2: How can it be proved that it is encapsulated, and what is the rate of encapsulation, is there any hemp oil that cannot loaded into this drug delivery system?

A nanoemulsion is a colloidal system formed by the dispersion in aqueous media of nanosized oil droplets thanks to the presence of a surfactant. The surfactant surrounds the oil droplet surface by forming a monolayer that prevents agglomeration and coalescence, thereby stabilizing the nanoemulsion. All the amount of the oil used in the preparation is supposed to form the droplets of the nanoemulsion, due to its practical insolubility in water. Therefore, the rate of oil encapsulation in nanoemulsions is much higher than 99%. All essential oils obtained from hemp or other plants are expected to be encapsulated in nanoemulsions at a rate much higher than 99%, as suggested by the available literature.

3: About the size, Zeta and PDI, Is there any significant difference between them, please mark it out.

The performed ANOVA followed by Sidak's multiple comparisons test evidenced statistical differences in the measured values of Z-average and PDI over time between nanoemulsions prepared with the two different hemp EOs. No statistical differences were found between PDI values at the time points T60 and T120. Despite the results of the statistical test, the differences highlighted are expected not to have practical relevance for their applications since all Z-average (range 110-165 nm) and PDI values (range 0.24-0.28) are similar.

4: In the cell cytotoxicological results, you were using many different types of cells. Could you please explain in detail why the toxicity is reduced after making nanoemulsion? In your article, you said it attributed to the presence of polysorbate 80. Could you please add some date or published paper to prove it? Another question, what is the crtl? If it is a control group, why does it also have toxicity?

The reduced toxicity of NEs in comparison to EOs, as evidenced from the calculated IC50 values, is only apparent and related to the composition of the NE, in which the oil does not represent the total weight (or volume) of the system but, only a percentage. In the case of NEs tested in this study, the oil represents only 5% or 6% w/w with respect to the total composition. Therefore, for NEs, the toxicity is affected by both the percentage (or actually, the content) of the EO but also by the presence of the other components as the surfactant (Polysorbate 80). Therefore, for formulations as NEs, it is necessary to normalize the calculated IC50 values according to the percentage of EO (normalized values reported in brackets in Table 2) to assess the effect of the formulation on EO toxicity. Indeed, after the normalization, the determined IC50 values for both NEs were like those calculated from the pure EOs. This means that the formulation of EO as nanoemulsion does not negatively affect the cytotoxicity of EOs. Moreover, it is also relevant to take into consideration the possible contribution in cytotoxicity of the other components of NE as the surfactant. Indeed, many surfactants being amphiphilic molecules, display an intrinsic cytotoxic effect due to their ability to partition inside the biological membranes. In this work, Polysorbate 80 was selected as a surfactant since it has the best performances in stabilizing NEs and it has also a low toxicity on cells, as reported by many previous studies. Control groups are NEs without hemp EOs and prepared according to the same procedure. This information has now been added in the method section (paragraph 4.4) For this reason, the controls showed a residual cytotoxicity that is much lower than that observed for the NEs also containing hemp EOs.

5: About the Figure 3 Hacat IL-1β group. EO and NE group cause the high level of IL-1β, how to explain it?

The test for quantifying the IL-1β expression on HaCAT cell lines has been repeated in three independent experiments and mean values obtained are reported in Figure 3. The authors are conscious that it is a paradox result in comparison to all other cytokines expressions on the investigated cell lines, in which cytokines expression levels are lower than those induced by ETO. The reported values in Figure 3 for EO Carmagnola CS, NE Carmagnola CS, and EO Uso 31 could suggest that skin cell lines are more sensitive to EOs than lung cells in relation to IL1-beta expression. This aspect should be taken into consideration in further studies on cytokines-related skin diseases.

6: The workload of this article is a bit small, could you add some more data?

Being a preliminary work about the evaluation of the cytotoxicological profile and cytokines expression level in three selected cell lines as human keratinocytes (HaCaT), human fibroblasts (NHF A12), and human bronchial cell (HBEpC), the authors have provided sufficient data to support the aim of the work stated in the introduction that is the comparison between the safety profile of pure hemp EOs and their NEs in order to assess the effect of nanoencapsulation on cytotoxicity and inflammation markers using MTT cell toxicity assay and gene expression studies. In addition, the manuscript also reports the extraction and characterization of two hemp EOs and their formulations into NEs. However, we agree with the reviewer that the issue of EOs formulation should be studied in more detail, since the interest on the use of cannabis products for cosmetic and medical uses.

Reviewer 2 Report

The authors describe cytotoxicity of nano-hemo using cell cultures. They provide evidence the nano-hemp has less toxicity than bulk-hemp.

I recommend minor corrections because of some questions I have about the results. Please split discussion in separate paragraphs.

Author Response

Reviewer 2

The authors describe the cytotoxicity of nano-hemp using cell cultures. They provide evidence the nano-hemp has less toxicity than bulk-hemp.

I recommend minor corrections because of some questions I have about the results.

The authors thank the reviewer for the positive general comment about the manuscript.

Please split discussion in separate paragraphs.

As suggested by the reviewer, the discussion section has been split into separate paragraphs.

Can you include a chromatogram?

The request of the reviewer is unclear since this comment was placed as a note of Table 2 dealing with the IC50 values of EOs and NEs calculated from the MTT assay performed on the three selected cell lines. The IC50 values are not derived from chromatograms but from the data presented in Figure 2.

What are these two values. Explain.

As already explained in the text, IC50 values of NEs after normalization by the percentage of EO contained in the formulation are reported in the brackets. For the sake of clarity, this information has been also added as a footnote in Table 2. 

Round 2

Reviewer 1 Report

This manuscript has been sufficiently improved. Thanks